# Hyperbolic exciton polaritons in a van der Waals magnet

Francesco L. Ruta [1,2,12] ✉, Shuai Zhang [1,12] ✉, Yinming Shao [1,12], Samuel L. Moore [1], Swagata Acharya[3], Zhiyuan Sun [1], Siyuan Qiu[1], Johannes Geurs[1,4], Brian S. Y. Kim [1,5], Matthew Fu[1], Daniel G. Chica[6], Dimitar Pashov [7], Xiaodong Xu[8,9], Di Xiao [8,9], Milan Delor[6], X-Y. Zhu [6], Andrew J. Millis [1,10], Xavier Roy [6], James C. Hone [5], Cory R. Dean [1], Mikhail I. Katsnelson [11], Mark van Schilfgaarde[3] & D. N. Basov [1] ✉

Exciton polaritons are quasiparticles of photons coupled strongly to bound electron-hole pairs, manifesting as an anti-crossing light dispersion near an exciton resonance. Highly anisotropic semiconductors with opposite-signed permittivities along different crystal axes are predicted to host exotic modes inside the anti-crossing called hyperbolic exciton polaritons (HEPs), which confine light subdiffractionally with enhanced density of states. Here, we show observational evidence of steady-state HEPs in the van der Waals magnet chromium sulfide bromide (CrSBr) using a cryogenic near-infrared near-field microscope. At low temperatures, in the magnetically-ordered state, anisotropic exciton resonances sharpen, driving the permittivity negative along one crystal axis and enabling HEP propagation. We characterize HEP momentum and losses in CrSBr, also demonstrating coupling to excitonic sidebands and enhancement by magnetic order: which boosts exciton spectral weight via wavefunction delocalization. Our findings open new pathways to nanoscale manipulation of excitons and light, including routes to magnetic, nonlocal, and quantum polaritonics.

Light interaction with excitons, bound states of electrons and holes in crystals, is a rich subject that has been studied extensively in semiconductor quantum wells[1–11] and – more recently – in van der Waals (vdW) semiconductors[12–21]. When excitons and light couple strongly, they form bosonic quasiparticles termed exciton polaritons[22]. Exciton polaritons display intriguing physics: they have been made into Bose-Einstein condensates[3,4] demonstrating superfluidity[5,6] with quantized vortices[7,8]. Furthermore, exciton polaritons have been used to realize

photonic topological insulators[9], low-threshold switching devices[10], lasing without population inversion[11], etc. Hyperbolic vdW materials, where permittivities have opposite signs along different crystal axes, are predicted to host exotic kinds of exciton polaritons called hyperbolic exciton polaritons (HEPs)[23,24]. HEPs may confine light to deeper subdiffractional wavelengths and provide enhanced polaritonic density of states relative to conventional exciton polaritons[20,21]. Hyperbolic polaritons can be composed of any polar excitation: they have been

[1]Department of Physics, Columbia University, New York, NY, USA. [2]Department of Applied Physics and Applied Mathematics, Columbia University, New York, NY, USA. [3]National Renewable Energy Laboratory, Golden, CO, USA. [4]Columbia Nano Initiative, Columbia University, New York, NY, USA. [5]Department of Mechanical Engineering, Columbia University, New York, NY, USA. [6]Department of Chemistry, Columbia University, New York, NY, USA. [7]Theory and Simulation of Condensed Matter, King's College London, London, UK. [8]Department of Physics, University of Washington, Seattle, WA, USA. [9]Department of Materials Science and Engineering, University of Washington, Seattle, WA, USA. [10]Center for Computational Quantum Physics, Flatiron Institute, New York, NY, USA. [11]Institute for Molecules and Materials, Radboud University, Nijmegen, Netherlands. [12]These authors contributed equally: Francesco L. Ruta, Shuai Zhang, Yinming Shao. ✉e-mail: f.ruta@columbia.edu; sz2822@columbia.edu; db3056@columbia.edu

observed with phonons[25,26], plasmons[27], and even transient excitonic transitions[28], but their observation with steady-state excitons remains challenging. HEP confinement can reach length scales comparable to the exciton Bohr radius, leading to unique nonlocal and quantum effects[23,29]. An experimental realization of HEPs will thus open new pathways to manipulate excitons and light at the nanoscale.

Exciton polaritons are commonly achieved within Fabry-Pérot microcavities, where highly-reflective mirrors on either side of a semiconducting well ensure observable coupling of excitons and photons. Energy oscillates between trapped photons and excitons (Rabi oscillations), leading to Rabi splitting of mode frequencies and coalescence of scattering rates[1,2]. Slabs of vdW semiconductors without a closed cavity can themselves serve as low-quality resonators that support propagating waveguide modes interacting with excitons[14], in a process sometimes referred to as self-hybridization[15,21]. This geometry lends itself to scanning probe nano-optical techniques since the material surface is exposed, permitting direct nano-imaging of exciton polaritons (Fig. 1a). At room temperature, waveguide modes reveal negative phase velocity or "backbending" dispersion and increased dissipation around exciton resonances[16–18]. Upon cooling, exciton resonances sharpen, causing an anti-crossing in the waveguide or cavity mode dispersion[19]. In hyperbolic materials, we will show HEPs appear inside the anti-crossing with multimode dispersion.

HEP imaging is technically challenging since light must acquire a large momentum to couple to HEPs, and exciton resonances must be strong enough to drive the permittivity to negative values while keeping damping low – requiring cryogenic temperatures in the case of most vdW materials[30]. In this work, we developed a cryogenic near-infrared near-field microscope to satisfy these experimental requirements and present direct images of steady-state HEPs in the vdW semiconductor chromium sulfide bromide (CrSBr). Excitons in CrSBr have large oscillator strength and small scattering rate even in bulk crystals, enabling observation of exciton polaritons without the need for a closed cavity[21]. Using our home-built nano-optics apparatus, we establish the existence of HEPs through a combination of energy-, temperature- and thickness-dependent measurements. Furthermore, CrSBr is known to be an A-type antiferromagnet (AFM), where individual ferromagnetic vdW layers order antiferromagnetically below the bulk Néel temperature $T_N = 132K$[31,32]. Excitons in CrSBr have been found to couple to magnetic order[33,34] and additional unidentified optical transitions appear below $T_N$[20,35]. We observe HEPs in CrSBr coupling to these optical transitions, indicating that these transitions persist in the high-momentum response. Finally, we note that the onset of hyperbolicity is concurrent with the appearance of intralayer ferromagnetic order at 160K[32,36], suggesting magneto-electronic coupling may be at play. Calculations within a self-consistent excitonic-vertex-corrected many-body perturbative approach (QS$G\hat{W}$)[37] reveal exciton delocalization upon magnetic ordering, which contributes to increased exciton spectral weight and robust hyperbolicity in CrSBr.

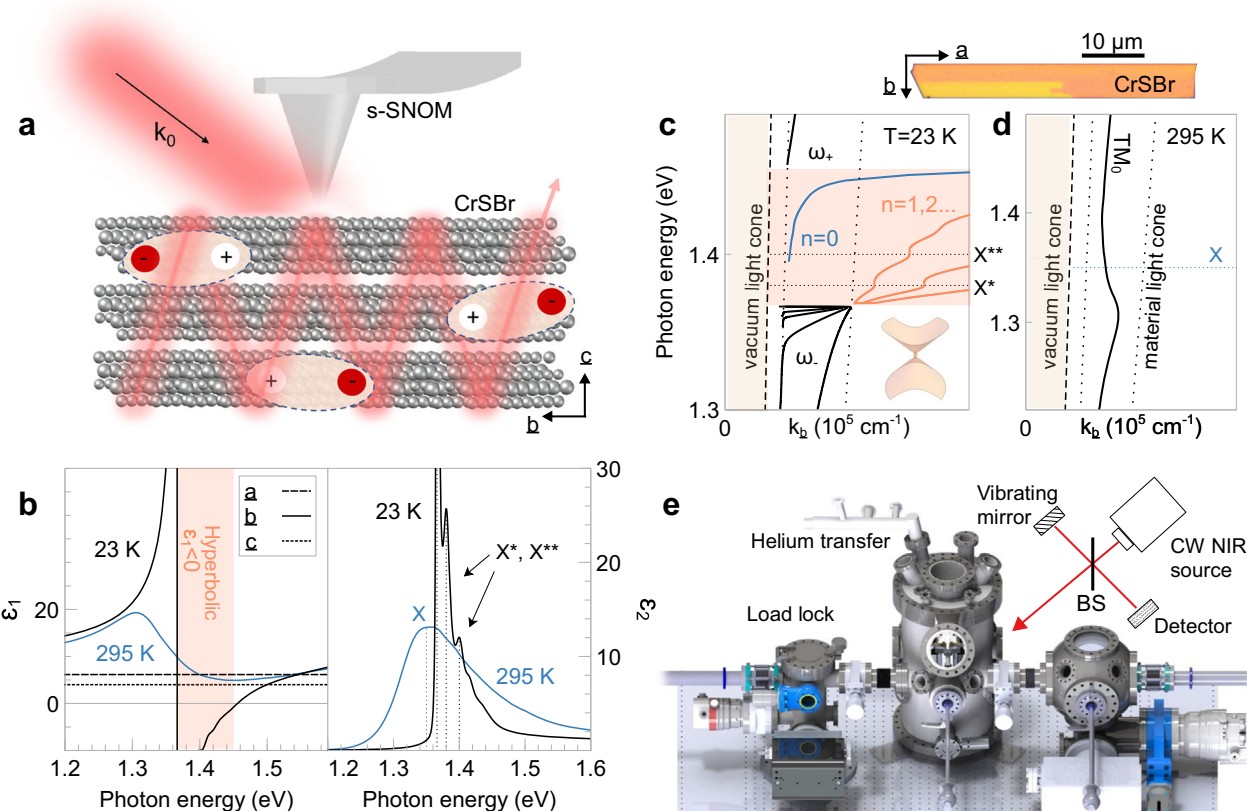

**Fig. 1 | Hyperbolicity in bulk chromium sulfide bromide (CrSBr). a** Schematic of metallized tip of a scattering-type scanning near-field optical microscope (s-SNOM) illuminated by free-space light with momentum $k_0$. The tip launches waveguide modes that couple to excitons (shown as electron (−)-hole (+) pairs) in a semiconducting van der Waals slab. **b** Experimental complex bulk CrSBr dielectric functions $\varepsilon = \varepsilon_1 + i\varepsilon_2$ at 295K (blue) and 23K (black) highlighting the temperature dependence of the exciton resonance (X). CrSBr has a hyperbolic band (orange region) with negative $\varepsilon_1$ only along the $b$-axis at low temperature. Additional resonances (X$^*$, X$^{**}$) following X are observed. **c** Transverse magnetic waveguide dispersions split (TM$_i \rightarrow \omega_+$ and $\omega_-$) about the hyperbolic band; fundamental ($n = 0$) and higher order ($n = 1,2,...$) hyperbolic exciton polaritons (HEPs) appear inside the gap at low temperature and couple to X$^*$ and X$^{**}$. **d** At room temperature, there is no hyperbolicity and the TM$_0$ waveguide mode only experiences gentle backbending at X between the SiO$_2$ and CrSBr light cones (dotted lines). **e** Beam path from a continuous-wave (CW) near-infrared (NIR) source through a beam-splitter (BS) going to a cryogenic s-SNOM, detector, and vibrating mirror.

## Results

### Hyperbolic exciton polaritons in CrSBr

We first extract in-plane optical constants of bulk CrSBr at different temperatures from broadband reflectance measurements by fitting a multilayer model (Methods). CrSBr is orthorhombic with *Pmmn* space group and lattice parameters $a = 0.4767$ nm, $b = 0.3506$ nm, and $c = 0.7965$ nm[31]; it has a diagonal biaxial dielectric tensor $\overleftrightarrow{\varepsilon} = \overleftrightarrow{\varepsilon}_1 + i\overleftrightarrow{\varepsilon}_2$ in the principal basis. The *b*-axis dielectric permittivity is dominated by exciton resonances below the semiconducting bandgap at 1.5eV[38]. In this work, we focus on the exciton resonance (X) peaked at 1.343eV (295K)-1.3675eV (20K). The X linewidth is broad at room temperature but narrows significantly at cryogenic temperatures (Fig. 1b), likely due to suppression of phonon-induced scattering[39]. Along the *a*- and *c*-axes, the dielectric functions are effectively constant in the near-infrared range ($\varepsilon^a = 12$ at 295K; $\varepsilon^c = 3.7$ at 20K, 4.5 at 50K, and 8 at 295K). *c*-axis values were extracted from near-field microscopy (Supplementary Figure 3). Because X appears only along the *b*-axis and is sharp enough at low temperature to allow for negative $\varepsilon_1$ values, we observe an energy band around 1.4eV (orange region in Fig. 1b) where the CrSBr isofrequency surface is hyperboloidal. Further, we note the existence of sidebands of X, labeled X* and X**, at 1.3830eV and 1.3935eV, respectively, in the low-temperature *b*-axis dielectric function (Fig. 1b, right panel).

Strong coupling between excitons and photons causes an anti-crossing of the waveguide mode dispersion at low temperature, and the highly anisotropic nature of the excitons in CrSBr also admits HEPs between the split waveguide modes (Fig. 1c). By contrast, at room temperature, X is broad and there is no hyperbolic band; we only expect a gentle backbending of the waveguide mode dispersion (Fig. 1d). HEPs can be either surface or bulk modes. Surface modes disperse from inside the material light cone ($\sqrt{\varepsilon^c}E = k\hbar c$) and have evanescent surface fields. Bulk modes, on the other hand, exist only to the right of the material light cone. In CrSBr, bulk modes have in-plane hyperbolic wavefronts with oscillatory out-of-plane electric fields propagating through the bulk like waveguide modes. The fundamental ($n = 0$) HEP mode is a surface mode inside the material light cone and becomes a bulk mode outside. On the other hand, higher-order HEP modes ($n = 1, 2, \ldots$) always exist as bulk modes (Supplementary Note 1). Furthermore, the sidebands noted earlier in Fig. 1b may couple to HEPs. Indeed, the backbending features in the HEP dispersion (Fig. 1c) occur at X* and X** energies.

### Nano-imaging experiments

To access HEPs experimentally, we performed cryogenic scattering-type scanning near-field optical microscopy (s-SNOM) on a home-built system illuminated by a tunable continuous-wave near-infrared laser (Fig. 1e). We also use a commercial s-SNOM system to characterize CrSBr waveguide modes at room temperature (Methods). In s-SNOM experiments, a tapping metallized tip is illuminated by laser light and the backscattered signal is demodulated at the higher harmonics of the tip-tapping frequency. The tip can launch modes, including high-momentum polaritons that cannot be excited with free-space light, which propagate to sample edges, transmit or reflect, and interfere with backscattered light from the tip. As the tip scans, interference fringes appear, corresponding to various light modes inside the material (e.g. waveguide modes and polaritons) or air modes, which are free-space standing waves from interfering scattering at the tip and sample edge that disperse like the vacuum light cone ($E = k\hbar c$).

We verify at room temperature that excitons and photons couple directionally in CrSBr. Figure 2 shows select room-temperature s-SNOM images and line profiles from corners of CrSBr crystals at photon energies in the range $1.26 - 1.55$eV. Fringes in Fig. 2a, b are superpositions of the fundamental transverse magnetic wave-guide mode (TM$_0$) and air mode interference fringes. Near-field amplitude line profiles along the *b*-axis (Fig. 2c) show a region of negative or backbending dispersion near X (follow dashed lines). On the other hand, line profiles along the *a*-axis (Fig. 2d) maintain a positive linear dispersion throughout the investigated energy range, indicating that *a*-axis TM$_0$ modes do not interact with excitons. In Fig. 2e, after applying a standard geometrical correction (Supplementary Note 3), Fourier peaks of both *a*- and *b*-axis profiles are overlaid on the Im $r_p$ loss function based on the dielectric functions obtained from far-field reflectometry. Fast Fourier transforms (FFTs) of line profiles in Figs. 2c, d can be found in Supplementary Fig. 7. We note excellent agreement between the calculated dispersion and momenta extracted from near-field profiles, affirming that exciton-photon coupling occurs only about the *b*-axis. Furthermore, within our calculation, the *b*-axis backbending in real $(\omega, k)$ space (Fig. 2e) corresponds to a Rabi splitting in complex-$\omega$ electrodynamics of $\Delta\omega = 226$meV (Supplementary Fig. 6c, Supplementary Note 2).

Next, we perform near-field nano-imaging of CrSBr at cryogenic temperatures. The waveguide mode dispersion now splits about the hyperbolic band with a complex-$\omega$ Rabi splitting energy of $\Delta\omega = 163$meV at 20K (Supplementary Fig. 6d). $\Delta\omega$ is smaller at 20K than at room temperature because of the reduced $\varepsilon^c$ (Supplementary Fig. 3), which was overlooked by other studies[20]. We expect HEPs to appear within the splitting. Figure 3a shows s-SNOM images at $E = 1.378$eV of a 200nm CrSBr crystal at $T = 295$K and 50K in the same region. Both images are normalized to the substrate and plotted on the same color scale. The low-temperature image is noticeably brighter, consistent with a negative $\varepsilon_1$ and the appearance of a hyperbolic band. Moreover, additional subdiffractional fringes with different periodicity appear in the 50K image (black arrows). After a geometrical correction (Supplementary Note 3), their corresponding momentum is $\sim 1.71 \times 10^5$ cm$^{-1}$ at 20K; which is beyond the material light cone with $k(E = 1.376\text{eV}, T = 20\text{K}) \approx 1.34 \times 10^5$ cm$^{-1}$. This momentum is consistent with the $n = 1$ bulk HEP mode.

In real-space (Fig. 3b), HEP fringes (orange diamonds) are partially obscured by air modes (green circles). In FFT spectra (Fig. 3c), however, HEPs (orange diamond) and air modes (green circle) can be readily distinguished. We fit two Lorentzian lineshapes (dashed blue lines) to the measured FFT spectrum, decompose them, and show their corresponding real-space inverse FFTs in Fig. 3b. If only the HEP peak is filtered in, then the corresponding real-space fringes are shown in the bottom of Fig. 3b. We extract an HEP propagation length $L_p \approx 0.5\mu$m (Methods) with a confinement factor $k/k_0$ of $\sim 2.5$ at this energy. Note that $k/k_0$ can be even higher at the edge of the hyperbolic band and for thinner samples or higher-order modes. A two-dimensional FFT filter can similarly be used to remove air mode fringes from Fig. 3a. The filtered near-field image at 50K is shown in Fig. 3d. The filtered regions of Fourier space are circled in red in the bottom inset. The HEP fringes are now unobscured and their averaged line profile (black line) is comparable to the one-dimensional Fourier-filtered HEP line profile in Fig. 3b.

Line profiles along the white dashed *b*-axis line in Fig. 3a are shown in Fig. 3e at $T = 295, 100, 50$, and 20K in black, gray, cyan, and purple, respectively. At room temperature, we observe TM$_0$ waveguide modes and air modes with similar periodicities. At 100K, TM$_0$ modes disappear as the waveguide mode dispersion splits and only air modes remain. New peaks appear between the air mode fringes at even lower temperatures, changing position from 50 to 20K (follow black dashed lines). Calculations using far-field optical constants suggest that the HEP wavelength should increase with decreasing temperature (Supplementary Fig. 9a) as $\varepsilon_1$ becomes more negative – in agreement with the experimental line profiles. Figure 3f shows a 50K near-field image outside the hyperbolic band, at $E = 1.304$eV, with corresponding FFT spectrum in Fig. 3g. Figure 3f looks qualitatively different from Fig. 3a at 50K. We now see the TM$_0$ waveguide mode since the probe energy is far from the anti-crossing. Unlike the HEP in Fig. 3c, the corrected momentum of the TM$_0$ peak in Fig. 3g is $\sim 1.2 \times 10^5$ cm$^{-1}$, less than the

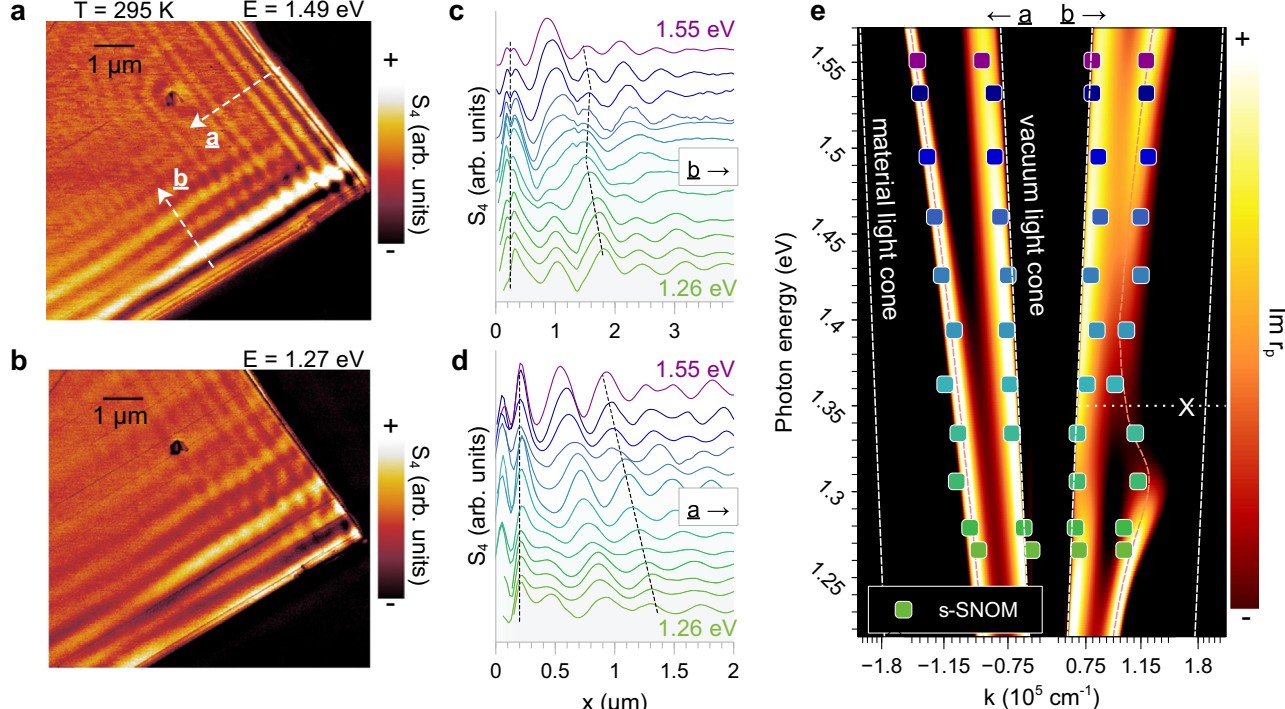

**Fig. 2 | Room temperature nano-imaging of CrSBr waveguide modes.** Room temperature near-field images at, (**a**) $E = 1.49$ eV and, (**b**) $E = 1.27$ eV showing interference fringes along $a$- and $b$-axes of a 117nm CrSBr microcrystal. **c**, line profiles along the white dashed $b$-axis line in (**a**) at probe energies from $E = 1.26 - 1.55$ eV. A region of negative dispersion (black dashed line) is observed near the exciton resonance (X). **d** Line profiles along the $a$-axis line in (**a**) have a positive linear dispersion (black dashed line) throughout the $E = 1.26 - 1.55$ eV energy range. **e** Geometry-corrected Fourier peaks of line profiles in (**c**, **d**) overlaid on the Im $r_p$ loss function calculated using optical constants from far-field spectroscopy. s-SNOM data are consistent with the vacuum light cone and waveguide mode dispersions. The waveguide mode dispersion backbends near X only along the $b$-axis.

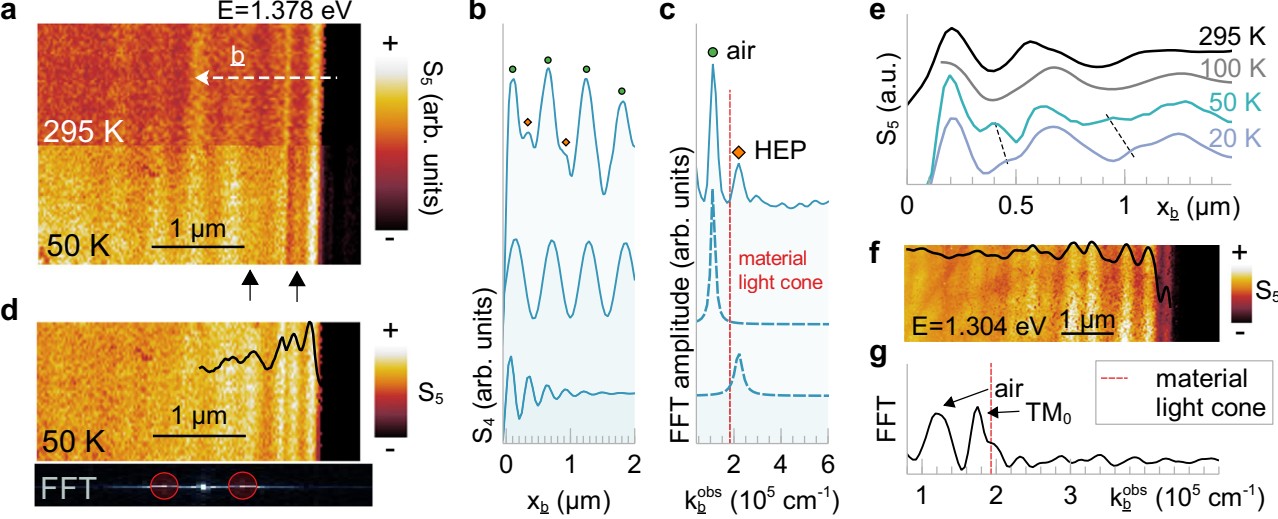

**Fig. 3 | Low temperature nano-imaging of hyperbolic exciton polaritons in CrSBr. a** Collocated $E = 1.378$ eV near-field images of a 200nm CrSBr microcrystal at $T = 295$K and 50K. Images are normalized to the area-averaged substrate signal and displayed on the same color scale. CrSBr becomes significantly brighter at 50K and new subdiffractional fringes appear (black arrows) corresponding to the suspected hyperbolic exciton polariton (HEP). **b** $b$-axis near-field line profile and, (**c**) corresponding fast Fourier transform (FFT) revealing observed momenta $k_b^{obs}$ of the air mode (green circle) and HEP (orange diamond). Decomposed air mode and HEP FFT peaks (blue dashed lines) with corresponding real-space profiles shown in (**b**). **d** Near-field image at 50K with air modes removed via 2D Fourier filtering. The bottom inset indicates the filtered regions of the 2D FFT. **e** $b$-axis line profiles along the white dashed line in (**a**) at $T = 295$K, 100K, 50K, and 20K. The subdiffractional fringes only appear in low-temperature profiles and redshift with decreasing temperature (dashed lines). **f** 50K near-field image outside hyperbolic band and averaged line profile (black line) with, (**g**) corresponding FFT reveal $TM_0$ mode inside the material light cone (red dashed line).

momentum of the material light cone (red dashed line). Also, the $TM_0$ mode is much less lossy than the HEP, as seen by either comparing propagation lengths or FFT peak intensities relative to air modes.

To further confirm our assignment of HEPs, we investigated another CrSBr crystal of different thickness. Figure 4a shows a near-field image at the corner of a 107nm CrSBr crystal at $T = 20$K and $E = 1.376$eV. We again see a subdiffractional fringe propagating along the $b$-axis. No subdiffractional fringe was observed along the $a$-axis – as expected from the in-plane hyperbolic isofrequency contour (left panel). The large anisotropy between observed $a$- and $b$-axis air modes, on the other hand, is due to the different interference paths along the two directions; which are also different from those in Fig. 2 (Supplementary Fig. 8). In Fig. 4b, the averaged $b$-axis line profile in the 107nm sample (blue) is compared to a line profile from the 200nm sample (purple) at the same temperature and photon energy. The fringe wavelength shortens with decreasing thickness, which is consistent with the well-established property that hyperbolic modes have higher momenta in thinner samples[25,26]. By contrast, waveguide modes follow the opposite trend (Supplementary Fig. 9b): allowing us to distinguish HEPs from waveguide modes and conventional exciton polaritons.

## Discussion

Next, we compare HEP momenta to calculations and assess the experimental evidence for coupling between HEPs and exciton sidebands. Figure 4c shows experimental HEP momenta (blue squares) from near-field data taken on a microcrystal with thickness $d = 107$nm at $T = 20$K and $E = 1.367 - 1.39$eV. FFTs of line profiles can be found in Supplementary Fig. 7. Error bars represent full-width half-maxima of FFT peaks. Data are overlaid on the $Im r_p$ loss function (maxima in orange) and calculated dispersion (white line). Note that $max Im r_p$ is not always a good indicator of poles for lossy modes near the light cone (Supplementary Note 1). Our data agree with the calculated dispersion (white line) and support the existence of backbending near $X^*$ - indicating that HEPs couple to exciton sidebands. Furthermore, near-field images taken with $E > 1.381$eV do not have noticeable HEP fringes, suggesting that $X^*$ may be enhancing polariton dissipation. Figures 4d,

e show near-field images at $E = 1.387$eV and 1.376eV, respectively. The black arrow indicates the HEP fringe in the 1.376eV image, which is absent in the 1.387eV image. Figure 4f shows calculated HEP propagation lengths with and without $X^*$. The $X^*$ sideband causes a significant reduction in propagation length at higher energies, which explains the absence of HEP fringes in Fig. 4d.

Recall that bulk CrSBr is an A-type AFM below $T_N = 132$K. Intralayer ferromagnetic interactions aligning in-plane spins of Cr orbitals (Fig. 5a) actually appear at a slightly higher temperature of $T_C = 160$K[32]. Between $T_N$ and $T_C$, short-range ferromagnetic domains form[36,40] before giving way to AFM order when interlayer magnetic coupling dominates below $T_N$ (Fig. 5b). In Fig. 5c, experimental $b$-axis $\varepsilon_1$ from far-field spectroscopy is shown near $T_N$ and $T_C$. The hyperbolic band of CrSBr caused by X (white region, $\varepsilon_1 < 0$) first emerges immediately below $T_C$ and broadens with decreasing temperature. In Fig. 5d, we plot the experimental spectral weight (black crosses) of X (Equation M4) as a function of temperature. All optical conductivities are shown in Supplementary Fig. 11. The spectral weight increases rapidly below $T_C$, then plateaus below $T_N$. Similar behavior has been observed in other transition metal magnets and was attributed to magneto-electronic coupling[41–43]. QS$G\hat{W}$ calculations on CrSBr (Methods) indeed predict a significant increase in exciton oscillator strength going from paramagnetic (PM) to AFM states (gray lines) and show remarkable agreement with angle-resolved photoemission spectroscopy[44]. Electrons taking part in exciton formation gain kinetic energy and become more dispersive along the Γ-Y direction and so less localized in the AFM phase (Fig. 5e). Considering electrons hopping in the crystal lattice roughly as atomic transitions, such that hopping is forbidden between atoms of antiparallel spin, then electrons should indeed be more itinerant in ferromagnetically polarized monolayers than in a PM lattice with random spins. Simultaneously, the onsite $d$-$d$ components of the exciton wavefunction are reduced in the ordered AFM phase (Fig. 5f), leading to a larger oscillator strength[45,46]. Together with reduced scattering rates at low temperature, the increased spectral weight from magnetically-induced exciton delocalization allows for robust HEPs in CrSBr.

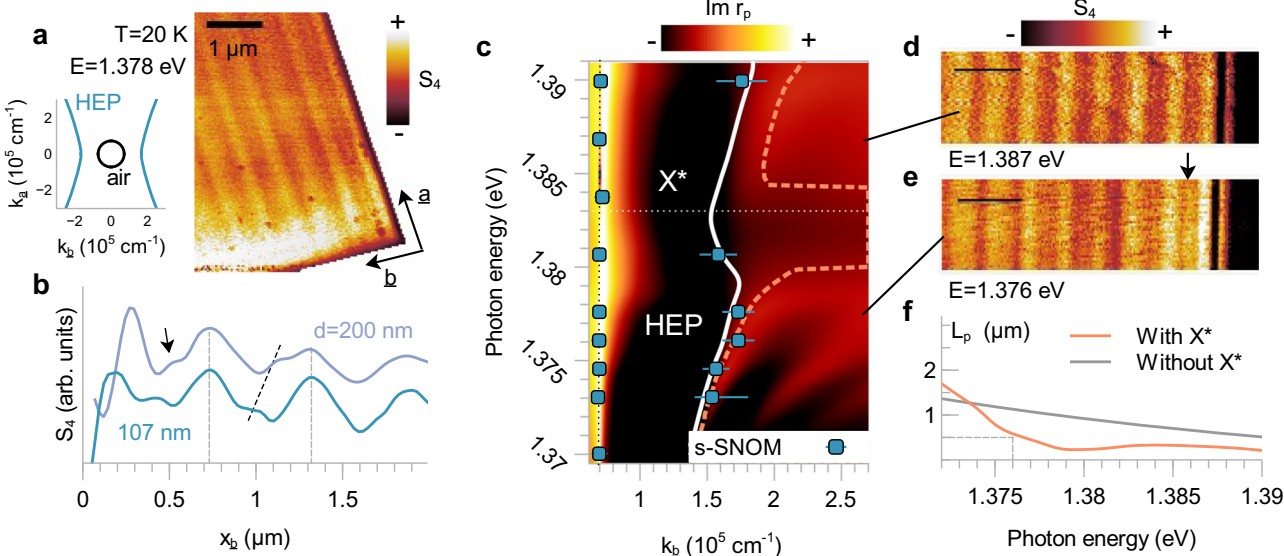

**Fig. 4 | Thickness dependence and dispersion of hyperbolic exciton polaritons.** **a** $T = 20$K near-field image of the corner of a 107nm CrSBr microcrystal at $E = 1.376$eV. The hyperbolic exciton polariton (HEP) fringe appears only along the $b$-axis as expected from its hyperbolic isofrequency contour (left inset). **b** Averaged $b$-axis line profile (blue) compared to a line profile at the same $T$ and $E$ on a 200nm crystal (purple). Air mode wavelengths do not change with thickness (aligned by vertical dashed lines) while the HEP fringe (black arrow) blueshifts with decreasing

thickness (diagonal dashed line). **c** Experimental HEP and air mode momenta at various energies overlaid on 20K $Im r_p$ loss function (dashed orange line corresponds to $max Im r_p$). $X^*$ sideband causes backbending of HEP dispersion (white line). **d** HEP does not appear in near-field image at $E = 1.387$eV, but, (**e**) appears in image at $E = 1.376$eV (black arrow). Scale bars are 1μm. **f** Theoretical propagation length $L_p$ with and without $X^*$ in orange and black, respectively. $X^*$ causes significant $L_p$ reduction.

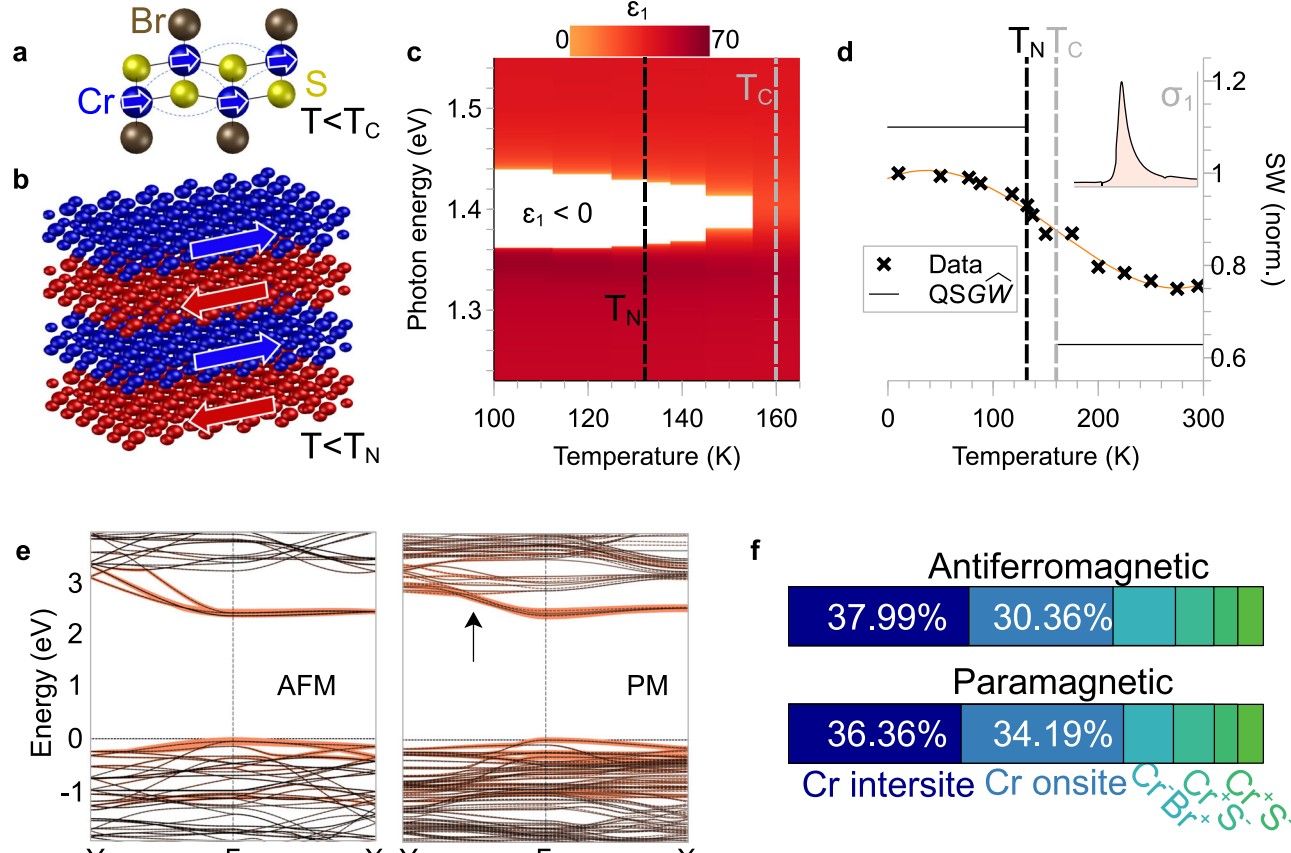

**Fig. 5 | Onset of hyperbolicity and magnetic order. a** Schematic of intralayer ferromagnetic correlations (dotted lines) aligning in-plane Cr spins (arrows) below a critical temperature $T_C = 160K$. **b** Interlayer correlations below the bulk Néel temperature ($T_N = 132K$) order CrSBr into an A-type antiferromagnet (AFM) with ferromagnetic van der Waals layers of alternating spin (arrows). **c** Real *b*-axis permittivity $\varepsilon_1$ near $T_N$ and $T_C$ from fits in Supplementary Fig. 11. CrSBr becomes hyperbolic, $\varepsilon_1 < 0$ (white), below $T_C$; and the hyperbolic band broadens with decreasing temperature. **d** Measured spectral weight (black crosses) of the CrSBr exciton (inset shows integral of real optical conductivity $\sigma_1$) increases rapidly with the onset of magnetic order at $T_C$ and plateaus below $T_N$. Orange line is a guide to the eye. Solid gray lines show QS$G\hat{W}$ theory predicting enhancement in exciton oscillator strength going from paramagnetic (PM) to AFM states. **e** QS$G\hat{W}$ electronic band structure in AFM (left) and PM (right) phases. Colored bands (orange) show the distribution of exciton spectral weight across various valence and conduction band states. Magnetic disordering in the PM phase leads to electron localization and reduced dispersion along Γ-Y (black arrow). **f** The probability of both electron (−) and hole (+) being on the same Cr site increases due to electron localization in PM phase.

In summary, we have observed HEPs in a steady-state near-field experiment. The temperature-, thickness-, and energy-dependence of subdiffractional fringes in our near-field experiments establish expected HEP properties. Further, we demonstrated coupling between HEPs and exciton sidebands in CrSBr, as evidenced by backbending and enhanced HEP losses above 1.381eV. Lastly, by measuring exciton spectral weight near magnetic transitions, we proposed that hyperbolicity in CrSBr is partly driven by magneto-electronic coupling. Future work may integrate CrSBr into an open cavity photonic crystal to improve HEP propagation lengths while still permitting direct imaging[47,48]. Improving quality may also enable imaging of the $n = 0$ surface mode – observations of surface exciton polaritons are rare[49,50] and near-field imaging could provide direct proof of their existence. Finally, measuring CrSBr in an s-SNOM capable of applying magnetic fields[51,52] would allow for studying the interplay between HEP propagation and magnetic order.

## Methods
### Single crystal growth
High quality $CrBr_3$ was synthesized from chromium powder (1.78g, 34.2mmol, 99.94%, −200 mesh, Alfa Aesar) and elemental bromine (8.41g, 52.6mmol, 99.99%, Sigma-Aldrich) with one end of the tube maintained at 1000 °C and the other at 50 °C via water bath to prevent the tube from exploding from bromine overpressure.

Chromium (0.174g, 3.35mmol), sulfur (0.196g, 6.11mmol, 99.9995%, Alfa Aesar), and $CrBr_3$ (0.803g, 2.75mmol) were then loaded into a 12.7mm outside-diameter, 10.5mm inside-diameter fused silica tube. The tube was evacuated to a pressure of ~30mTorr and flame sealed to a length of 20cm. The tube was placed into a computer-controlled, two-zone, tube furnace. The source side was heated to 850 °C in 24h, allowed to soak for 24h, heated to 950 °C in 12 h, allowed to soak for 48h, and then cooled to ambient temperature in 6 h. The sink side was heated to 950 °C in 24h, allowed to soak for 24h, heated to 850 °C in 12h, allowed to soak for 48 h, and then cooled to ambient temperature in 6h. The crystals were cleaned by soaking in 1mgmL$^{-1}$ of $CrCl_2$ (anhydrous, 99.9%, Strem Chemicals) aqueous solution for 1h at ambient temperature. After soaking, the solution was decanted and the crystals were thoroughly rinsed with deionized water and acetone. Residual sulfur residue was removed by washing with warm toluene.

### Stacking and transfer
At cryogenic temperatures, CrSBr contaminated s-SNOM tips during scanning. To overcome this, we encapsulated CrSBr with a thin (<10nm) hexagonal boron nitride (hBN) layer[53]. Exfoliated hBN is picked up using the standard dry-transfer technique. A polycarbonate (PC) film draped over a polydimethylsiloxane (PDMS) dome is gently pressed down onto the crystal at 130 °C. As the PC cools to 100 °C, the crystal is lifted from an $SiO_2/Si$ substrate and adheres to the polymer.

The hBN/PC film is then pressed onto an exfoliated CrSBr crystal following the same temperature protocol. To reduce CrSBr adhesion to the $SiO_2$, thereby ensuring hBN picks up the CrSBr, the $SiO_2$/Si chip is first treated with liquid 1-dodecanol at 160 °C for 5–10 min. Residual 1-dodecanol is then washed away with isopropyl alcohol (IPA) several times. Next, the hBN/CrSBr stack is transferred to a $SiO_2$/Si chip pre-patterned with ~200 μm × 200 μm gold pads. The PC temperature is slowly raised to 190 °C before delaminating from the PDMS dome onto the chip. The PC is then removed, leaving the finished stack, by rinsing with chloroform and IPA several times at 15 sec each step. Finally, the heterostructures were cleaned of any remaining polymer residue by contact mode scanning in a Bruker Dimension atomic force microscope.

## Fourier transform infrared spectroscopy

Reflectance spectra of CrSBr flakes were measured using a Bruker Hyperion 2000 microscope connected to a Bruker Vertex 80 V FTIR spectrometer. A tungsten halogen lamp was used as a light source covering a frequency range of 0.5 to $\sim 2.5$ eV. Linearly polarized light was focused on the sample using a 15$X$ objective and the aperture size was set to be smaller than the sample dimensions. Reflectance spectra in the visible range are normalized to an optically thick silver layer (170nm) on the substrate and recorded with a silicon detector.

Dielectric function fitting was performed with a multilayer model consisting of a CrSBr thin flake with thickness $d$ on top of a 90-nm-thick $SiO_2$ layer on a semi-infinite silicon substrate. The dielectric function of the $SiO_2$ layer is extracted from the measured reflectance using variational dielectric function (VDF) fitting in the RefFit package[54]. The $b$-axis dielectric response of CrSBr as a function of frequency was initially parameterized by a combination of Lorentz oscillators

$$\varepsilon(\omega) = \varepsilon_\infty + \frac{f}{\omega_0^2 - \omega^2 - i\gamma\omega} \tag{M1}$$

and Tauc-Lorentz oscillators

$$\varepsilon_2(\omega) = \sum_i \frac{A_i}{\omega} \frac{\omega_i \gamma_i (\omega - \omega_{g,i})^2}{(\omega^2 - \omega_i^2)^2 + \gamma_i^2 \omega^2} \Theta(\omega - \omega_{g,i}), \tag{M2}$$

$$\varepsilon_1(\omega) = \frac{2}{\pi} \int_0^\infty \frac{\omega' \varepsilon_2(\omega')}{\omega'^2 - \omega^2} d\omega' \tag{M3}$$

Here, $\omega_0$, $f$, and $\gamma$ are the oscillator frequency, oscillator strength, and linewidth of the Lorentz oscillator, respectively. $\varepsilon_\infty$ is the high-frequency dielectric permittivity. Tauc-Lorentz oscillators were used to better describe the asymmetric lineshape of the excitonic resonances in CrSBr. In Equation M2, $\omega_i$, $A_i$, $\omega_{g,i}$, and $\gamma_i$ describe the center frequency, height, onset frequency, and width of the Tauc-Lorentz oscillator, respectively. $\Theta(x)$ is a step function and the real part of the dielectric function $\varepsilon_1(\omega)$ is obtained from Kramers-Kronig relations (Equation M3). Using the multi-oscillator model as a starting point (Supplementary Table 1), reflectance spectra for $b$-axis responses of CrSBr were further refined with VDF fitting in the RefFit package[54].

Spectral weight was defined in this work by the following expression:

$$SW = \frac{h^2 c}{\pi^2 \varepsilon_0} \int_{\omega_{min}}^{\omega_{max}} \sigma_1(\omega) d\omega \tag{M4}$$

where $\sigma_1(\omega) = \text{Re}(ic\omega\varepsilon_0(1 - \varepsilon(\omega))) = c\omega\varepsilon_0\varepsilon_2(\omega)$ is the real part of the optical conductivity. The integrated spectral range was from $1.24 - 1.55$ eV ($10,000 - 12,500$ cm$^{-1}$). Spectral weight values were normalized such that the $SW(T = 10$ K$) = 1$.

## Scanning near-field optical microscopy

For room temperature measurements, a Neaspec neaSNOM near-field microscope was used with an M Squared SolsTiS continuous-wave Ti-sapphire laser. PtIr-coated NanoWorld Arrow tips with 75kHz resonance frequencies were used. For low temperature measurements, we used a home-built cryogenic near-field microscope also with an M Squared SolsTiS. Ag-coated TERS tips (OMNI TERS-SNC-Ag) from AppNano were used in the cryo-system. The signal localized under the apex of the tip is isolated in the backscattered signal by demodulation at the 1st–5th tip tapping harmonics. Pseudoheterodyne detection was employed whereby backscattered light is interfered with light from a vibrating mirror and only signal from interference sidebands was collected[55]. Temperature measurements were performed with a silicon diode sensor (Lakeshore DT-670) fixed underneath the sample holder receiver.

The Fourier filtered image in Fig. 3d was obtained using the 2D fast Fourier transform (FFT) filtering tool in Gwyddion. Air mode peaks at positive and negative momenta were removed from the mask, as shown by the red circles in the bottom inset of Fig. 3d. One-dimensional line profiles (Fig. 3b), on the other hand, were Fourier transformed using SciPy.fft methods and sometimes processed with BaselineRemoval and gaussianfilter_1d methods. Fourier filtering was performed by fitting Lorentzian functions to FFT peaks and analytically taking their inverse Fourier transform:

$$S \propto \mathcal{F}^{-1}\left[ \left( \left(\hat{k} - \text{Re } k\right)^2 + (\text{Im } k)^2 \right)^{-1} \right] \propto e^{-i(\text{Re } k)x} e^{-(\text{Im } k)|x|} \tag{M5}$$

We see that the propagation length $L_p$, defined as the distance $x$ until the real-space amplitude drops to $1/e$ of its initial value, can thus be estimated from the half-width at half-maximum Im$k$ of the Lorentzian peak simply as $L_p = 1/\text{Im } k$.

## Ab initio quasiparticle GW theory

Electronic structure and excitonic properties are calculated within a self-consistent diagrammatic extension of quasiparticle self-consistent $GW$ theory (QS$GW$)[56] called QS$G\hat{W}$. In contrast to conventional $GW$ methods, QS$GW$ modifies the charge density and is justified by a variational principle[57]. On the other hand, QS$G\hat{W}$[37] iteratively updates $G$, the self-energy ($\Sigma$), and the screened Coulomb interaction ($W$) until all quantities self-consistently converge to desired accuracies. Within QS$G\hat{W}$, $W$ includes vertex corrections (ladder diagrams) by solving a Bethe–Salpeter equation. Crucially, our QS$G\hat{W}$ methods are fully self-consistent in both $\Sigma$ and the charge density[58]. Our results are thus parameter-free and have no starting point bias. QS$GW$ and QS$G\hat{W}$ cycles are iterated until the root mean square change in $\Sigma_0$ reaches $10^{-5}$ Ry. A $2 \times 2 \times 2$ supercell of the $Cr_2Br_2S_2$ formula unit with six atoms per unit cell was used (48 atoms total with 48 interstitial sites added to augment the basis with floating orbitals). In paramagnetic calculations, local spin orientations are arranged in a quasirandom configuration that mimics the most relevant radial correlation functions of a true random structure. An objective function composed of 480 pair and 384 triplet functions is minimized, following the approach of Ref. 59. The objective function contained 16 pairs and 24 triplets per Cr site. This corresponds to three shells of Cr neighbors whose length ranged between 0.41 and 0.71 lattice constants, and all triplets whose sum of lengths did not exceed 1.37 lattice constants. The two-particle excitonic Hamiltonian that is solved self-consistently to compute both $\Sigma$ and excitonic eigenvalues and eigenfunctions, contained 104 valence bands and 36 conduction bands.

## Data availability

Relevant data supporting the key findings of this study are available within the article and the Supplementary Information file. All raw data

generated during the current study are available from the corresponding authors upon request.

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

## Acknowledgements
This work is supported as part of Programmable Quantum Materials, an Energy Frontier Research Center funded by the U.S. Department of Energy (DOE), Office of Science, Basic Energy Sciences (BES), under award no. DE-SC0019443. D.N.B. is a Moore Investigator in Quantum Materials EPIQS no. 9455. Synthesis and structural characterization of the materials was supported in part by the Columbia MRSEC on Precision-Assembled Quantum Materials (PAQM) under award number DMR-2011738. S.A., D.P., and M.v.S. are supported by the Computational Chemical Sciences program within the U.S. DOE, Office of Science, BES, under award no. DE-AC36-08GO28308. The development of nano-optical imaging at Columbia is supported by DOE-BES DE-SC0018426. D.N.B. is Moore Investigator in Quantum Materials EPIQS GBMF9455. M.I.K. is supported by the ERC Synergy Grant, project 854843 FAS-TCORR (Ultrafast dynamics of correlated electrons in solids). This research used resources of the National Energy Research Scientific Computing Center, a DOE Office of Science user facility supported under award no. DE-AC02-05CH11231 using NERSC award BES-ERCAP0021783. The Flatiron Institute is a division of the Simons Foundation.

## Author contributions
F.L.R., S.Z., and Y.S. contributed equally. F.L.R., S.Z., and M.F. performed cryogenic near-field experiments. F.L.R. performed room-temperature near-field experiments. F.L.R. and S.Z. analyzed the near-field data. Far-field infrared spectroscopy and analysis was carried out by Y.S. and S.Q. S.M., J.G., and B.S.Y.K. fabricated and cleaned samples for cryogenic measurements with supervision from C.R.D. and J.C.H. D.G.C. synthesized CrSBr crystals with supervision from X.R. S.A., D.P., M.I.K, and M.S. performed QSG$\hat{W}$ calculations. F.L.R. and Z.S. performed electrodynamics calculations. X.X, D.X., M.D., X.Y.Z., and A.J.M. helped to interpret the results. D.N.B. coordinated and supervised the work. F.L.R. and D.N.B. wrote the manuscript with input from all coauthors.

## Competing interests
The authors declare no competing interests.
