## [Peer Review File · Nature Communications]

Hyperbolic exciton polaritons in a van der Waals magnetREVIEWER COMMENTS

Reviewer #1 (Remarks to the Author):

The authors studied hyperbolic exciton polaritons (HEPs) in CrSBr, a Van der Waals magnetic semiconductor both theoretically and experimentally. CrSBr is a novel material system that has drawn great attentions recently due to its interesting photon-exciton-magnetic coupling. This work offers a comprehensive description of the optical properties of the material as well as a rigorously calculation about the HEPs, and the cryogenic s-SNOM technique applied for the near field observation is quite advanced. However, the real highlight of the article—the experimental illustrations of the HEPs are not convincing, and the magnetic tuning of the polariton property is missing.

Specifically, the Hyperbolicity in anisotropic polariton systems has been widely reported, and the theories are quite well established. Therefore, a concrete experimental proof of the existence of hyperbolic polaritons is critical, which requires hyperbolic isofrequency contours in k-space or a hyperbolic propagation. In this work, the only evidence for HEP demonstration is a mode with shorter wavelength comparing with the bulk mode, which, in my opinion, is insufficient to prove the observation of HEP.

Besides, the observed experimental features shown in are quite blurred. For example, Figure 3(a) only shows one weak addition fringe in the bottom part, which is not convincing enough for demonstrating an additional mode. Moreover, basic characterizations of this mode is missing, such as its wavelength and its comparison with the theoretical calculation. Also, in Figure4(a) it is hard to tell if there is a fringe independent from the other and in Figure 4(b) as it is too weak compared with the waveguide modes.

Moreover, a critical part that is missing is the magnetic response of the HEP. As the authors also mentioned, this would be a unique and intriguing property of this material.

In summary, the authors worked on a novel and interesting material system, however, the evidence for the existence of HEPs in CrSBr provided in this article is not sufficient and the experimental results are preliminary. Considering that the direct experimental observations of the HEPs are the core of the article, the current work is quite premature and needs significant revision, therefore I cannot recommend its publication at this form. I can reconsider it if the sample and data quality were significantly enhanced and the above issues were properly addressed.

Reviewer #2 (Remarks to the Author):

The manuscript reports on the nanoimaging of hyperbolic exciton polaritons at low temperatures. In order to conduct their experimental studies, the authors have constructed a unique setup allowing one

to perform the near-field measurements in real space at low temperatures in the near-IR frequency range. Because of low temperatures, a strong exciton resonance is achieved in the vdW semiconductor chromium sulfide bromide, so that its dielectric permittivity becomes negative in one of the crystal directions. This enables propagation of exciton polaritons with hyperbolic dispersion that surpass the diffraction limit. Moreover, the authors observe that the hyperbolicity is concurrent with the appearance of intralayer ferromagnetic order so that magneto-electronic coupling may play a crucial role.

The manuscript is very well organized: the illustrations and messages are clear, the data is convincing. The reported studies can be relevant for both optics (particularly, nanoptical and polaritonic communities), material science and condensed matter. The manuscript is recommended for publication in its current form.

Reviewer #3 (Remarks to the Author):

In the manuscript “Hyperbolic exciton polaritons in a van der Waals magnet”, the authors reported an experimental study (applying a cryogenic near-field microscope) of steady-state hyperbolic exciton polaritons in the van der Waals semiconductor chromium sulfide bromide. The experimental results are well corroborated with theoretical calculations, and the manuscript is clearly written. As the observation of hyperbolic exciton polaritons will be important for manipulating both excitons and polaritons at the nanoscale, I will recommend this manuscript for publication in Nature Communications once the following comments will be addressed:

1. I have no problem with the novelty of this work. However, the authors need to improve details to make it easier for readers: (i) could the authors add the line corresponding to material light cone in the Fig.2e and Fig.3b; (ii) the scale bar is missing in Fig.3e; (iii) the units are missing in Fig.4a.
2. There are some steps in the calculated dispersion of high-order modes of hyperbolic exciton polaritons (as shown in the Fig.1c). Are there any physical insights?
3. Why the near-field images in Fig.3a shows a much lower signal-to-noise ratio (both the waveguide modes and HEP modes) than the ones in other figures?
4. How did the authors do the 2D FFT from Fig.3a to Fig.3c? They need to be careful since it is quite easy to introduce some false signals in some commercial software when they do 2D FFT.
5. In Fig.4d, the calculated propagation length gives a value of 1 μm , while the authors claim that the HEP propagates for $\sim 1 \mu\text{m}$ in Fig.3b. How did they extract the experimental results of propagation length?

We would like to thank all reviewers for their time and insightful comments. We believe the manuscript has been significantly improved thanks to their suggestions. In particular, we have revisited our data processing to further improve imaging and Fourier transform quality. We now detail the exact procedure in the Methods section. Figs. 2a and 2c are reprocessed and we have taken care to be more rigorous with our propagation length extraction and comparison to theory. We obtained new magneto-optical spectra on CrSBr (Fig. R2). Also, we attempted antenna launching experiments, but were unsuccessful due to the fragile silver tips that need to be used in near-infrared cryo-SNOM (Fig. R1). Finally, some minor fluency and presentation improvements were also made.

Reviewer #1 Response:

The authors studied hyperbolic exciton polaritons (HEPs) in CrSBr, a Van der Waals magnetic semiconductor both theoretically and experimentally. CrSBr is a novel material system that has drawn great attentions recently due to its interesting photon-exciton-magnetic coupling. This work offers a comprehensive description of the optical properties of the material as well as a rigorously calculation about the HEPs, and the cryogenic s-SNOM technique applied for the near field observation is quite advanced. However, the real highlight of the article—the experimental illustrations of the HEPs are not convincing, and the magnetic tuning of the polariton property is missing.

Thank you for the careful review and astute comments. Please find our responses below.

Specifically, the Hyperbolicity in anisotropic polariton systems has been widely reported, and the theories are quite well established. Therefore, a concrete experimental proof of the existence of hyperbolic polaritons is critical, which requires hyperbolic isofrequency contours in k-space or a hyperbolic propagation. In this work, the only evidence for HEP demonstration is a mode with shorter wavelength comparing with the bulk mode, which, in my opinion, is insufficient to prove the observation of HEP.

We presented several other points of converging evidence for hyperbolicity besides the subdiffractive wavelength of the HEP mode. We summarize below the evidence forcing the HEP interpretation as the only plausible interpretation:

- (i) HEP fringes only appear at low temperature (Fig. 3d), consistent with far-field spectroscopy indicating robust hyperbolicity.
- (ii) Fringes shift position between T=50 K and 20 K, consistent with calculations (Fig. S9)
- (iii) The HEP peak is subdiffractive only within the hyperbolic band (Fig. 3b). Outside the hyperbolic band (Fig. 3e), the peak re-enters the material light cone.
- (iv) HEP fringes shift to shorter wavelength with decreasing sample thickness (Fig. 4b). This is a property unique to hyperbolic modes. Conventional exciton polaritons and waveguide modes follow the opposite trend.
- (v) The observed HEP dispersion closely follows calculated energy dependence for the n=1 HEP (Fig. 4c).
- (vi) The observed HEP is nearly exactly as lossy as expected from calculations (Fig. 4d). Conventional exciton polaritons or waveguide modes should be much less dissipative.
- (vii) No similar subdiffractive fringe was observed along the a-axis (Fig. 4a), indicating in-plane hyperbolic anisotropy.

While the reviewer is correct that a hyperbolic wavefront or directional launching would be strong evidence for hyperbolicity, these data are ultimately redundant given the many other pieces of evidence we discussed above.

Moreover, obtaining such data requires development of antenna launchers for near-infrared nano-optics. Although launchers are well-established in mid-infrared nano-optics, no room temperature near-infrared experiments have successfully demonstrated antenna launching to our knowledge. This is likely because near-infrared/visible wavelengths are much shorter, thus requiring small launcher dimensions and abrupt edges/features at the single-digit nanometers length scales. These later structures are technically challenging to fabricate, calling for alternative experimental tests of hyperbolicity including i)-vii) above. Moreover, launchers will still produce strong air mode fringes, which are guaranteed to obscure hyperbolic wavefronts as our modeling shows. Thus, the wavefront images may not be as easily interpretable as the reviewer may expect from their experience with mid-infrared nano-optics. The requirement of cryogenic near-infrared nanoscopy brings additional challenges. Our attempts at imaging electron-beam-deposited launchers in UHV (Fig. R1) end with signal loss once the very soft silver tip needed for near-infrared alignment attempts to scan across the launcher. Although we attempted the experiment suggested by the reviewer, we have no presentable near-field images due to this major challenge with the silver tip.

Figure R1: Topography image of electron-beam-deposited gold launcher that causes near-field signal loss in UHV with silver tips.

Besides, the observed experimental features shown in are quite blurred. For example, Figure 3(a) only shows one weak addition fringe in the bottom part, which is not convincing enough for demonstrating an additional mode. Moreover, basic characterizations of this mode is missing, such as its wavelength and its comparison with the theoretical calculation. Also, in Figure 4(a) it is hard to tell if there is a fringe independent from the other and in Figure 4(b) as it is too weak compared with the waveguide modes.

The presence of air modes superposed with the HEP is unfortunately an unavoidable consequence of the shorter wavelength of near-infrared light. However, while the real-space image may not be immediately interpretable because of the air modes, we show that the Fourier transform (Fig. 3b) has two unambiguous and robust peaks – one of which must correspond to an HEP by the evidence discussed previously.

The requested comparison of experimental and theoretical momenta/wavelengths was already presented in Fig. 4c of the original submission. We added a sentence (pg. 5, end of first paragraph) to bring attention to this figure.

Moreover, a critical part that is missing is the magnetic response of the HEP. As the authors also mentioned, this would be a unique and intriguing property of this material.

Magneto-SNOM is a technique that has only this year been first demonstrated for THz and mid-infrared light (see Refs. 50-51). Adapting this technique for near-infrared experiments is a tall order that will present many years of challenges. To note one, mid-infrared magneto-SNOM requires an Akiyama probe, which is incompatible with near-infrared cryo-SNOM. Indeed, near-infrared cryo-SNOM only works with silver elephant-trunk tips in our experience. Not to mention, implementing a tunable magnet in our near-infrared cryo-system would require a complete rebuild.

Moreover, we expect the magnetic field dependence of HEPs to lead to mostly minor changes in the data. Far-field magneto-optical spectroscopy suggests that the CrSBr exciton will slightly red-shift in the ferromagnetic state (consistent with several previous reports) and the sidebands will be suppressed. We thus expect the hyperbolic window to shift and the HEP dispersion to change slightly upon switching to ferromagnetic order. See Fig. R2 below.

Figure R2: left panel, reflectance spectrum of CrSBr with ferromagnetic (blue) and antiferromagnetic order (orange). Right panel, extracted imaginary dielectric function shows slight peak-shift and suppression of sideband in ferromagnetic state.

In summary, the authors worked on a novel and interesting material system, however, the evidence for the existence of HEPs in CrSBr provided in this article is not sufficient and the experimental results are preliminary. Considering that the direct experimental observations of the HEPs are the core of the article, the current work is quite premature and needs significant revision, therefore I cannot recommend its publication at this form. I can reconsider it if the sample and data quality were significantly enhanced and the above issues were properly addressed.

We would like to emphasize that the advanced near-infrared cryo-SNOM technique applied here comes with several unique technical challenges beyond what the reviewer may have experienced with mid-infrared or THz s-SNOM. The required soft silver tips and shorter wavelength of near-infrared light make many previous cryo-SNOM techniques unfeasible. We feel that many of the reviewer's comments are holding our experiments to the standards of mid-infrared cryo-SNOM, which is a much more well-established (circa 2015) and technically simpler technique. To achieve comparable image quality to mid-infrared with air mode suppression, antenna launching, and magneto-optics in the near-infrared will likely need years of further development. We thus believe some suggestions are beyond the scope of our first pioneering demonstration of near-infrared cryo-SNOM.

Nonetheless, the evidence for HEPs presented in our manuscript is still very strong regardless of the technical difficulties of near-infrared cryo-SNOM. The seven converging pieces of evidence discussed above and in the manuscript force the HEP interpretation. There is no other plausible explanation for our data.

Reviewer #2 Response:

The manuscript reports on the nanoimaging of hyperbolic exciton polaritons at low temperatures. In order to conduct their experimental studies, the authors have constructed a unique setup allowing one to perform the near-field measurements in real space at low temperatures in the near-IR frequency range. Because of low temperatures, a strong exciton resonance is achieved in the vdW semiconductor chromium sulfide bromide, so that its dielectric permittivity becomes negative in one of the crystal directions. This enables propagation of exciton polaritons with hyperbolic dispersion that surpass the diffraction limit. Moreover, the authors observe that the hyperbolicity is concurrent with the appearance of intralayer ferromagnetic order so that magneto-electronic coupling may play a crucial role.

The manuscript is very well organized: the illustrations and messages are clear, the data is convincing. The reported studies can be relevant for both optics (particularly, nano-optical and polaritonic communities), material science and condensed matter. The manuscript is recommended for publication in its current form.

Thank you for the careful reading and positive feedback on our work!

Reviewer #3 Response:

In the manuscript “Hyperbolic exciton polaritons in a van der Waals magnet”, the authors reported an experimental study (applying a cryogenic near-field microscope) of steady-state hyperbolic exciton polaritons in the van der Waals semiconductor chromium sulfide bromide. The experimental results are well corroborated with theoretical calculations, and the manuscript is clearly written. As the observation of hyperbolic exciton polaritons will be important for manipulating both excitons and polaritons at the nanoscale, I will recommend this manuscript for publication in Nature Communications once the following comments will be addressed:

Thank you for careful review of our manuscript and the helpful comments. The reviewer’s comments are addressed below.

1. I have no problem with the novelty of this work. However, the authors need to improve details to make it easier for readers: (i) could the authors add the line corresponding to material light cone in the Fig.2e and Fig.3b; (ii) the scale bar is missing in Fig.3e; (iii) the units are missing in Fig.4a.

(i) The material light cone is now indicated in Figs. 2e and 3b.

(ii) A scale bar was added to Fig. 3e.

(iii) Units were added to Fig. 4a.

2. There are some steps in the calculated dispersion of high-order modes of hyperbolic exciton polaritons (as shown in the Fig.1c). Are there any physical insights?

The steps in the HEP dispersion correspond to “backbending” from coupling with excitonic sidebands X^* and X^{**} . Coupling to X^* was experimentally verified in Figs. 4c and 4d. This observed coupling tells us that the sidebands measured in $q=0$ far-field spectroscopy must also persist in the high- q optical response of CrSBr (pg. 3, lines 3-4). We added labels to Fig. 1c.

3. Why the near-field images in Fig.3a shows a much lower signal-to-noise ratio (both the waveguide modes and HEP modes) than the ones in other figures?

This is an effect of the colormap scaling. In Fig. 3a, we present two images (295K and 50K) with different baseline levels of near-field amplitude on the same color scale. There is a tradeoff between apparent signal-to-noise ratio and visibility of fringes in both images. In other images, we could optimize the color scaling to the corresponding image. We tried to redo the scaling to improve the quality slightly:

Figure R3: Left, old, and right, new rescaled version of Fig. 3a. The apparent image quality is improved. Note that labels are not shown here but are shown in the main text version.

4. How did the authors do the 2D FFT from Fig.3a to Fig.3c? They need to be careful since it is quite easy to introduce some false signals in some commercial software when they do 2D FFT.

We performed 2D and 1D FFT filtering in different software packages and obtained consistent results, which gave us confidence that we are not producing artefacts. 2D FFT filtering (Fig. 3c) was performed with Gwyddion and 1D FFT filtering (Fig. 3b) with SciPy. You may compare the line profiles in the bottom left of Fig. 3b (SciPy) and top right of Fig. 3c (Gwyddion). We added precise details of the filtering techniques to the Methods section:

The Fourier filtered image in Fig. 3c was obtained using the 2D fast Fourier transform (FFT) filtering tool in Gwyddion. Air mode peaks at positive and negative momenta were removed from the mask, as shown by the red circles in the bottom panel of Fig. 3c. One-dimensional line profiles, on the other hand, were Fourier transformed using SciPy.fft methods (Fig. 3b) and sometimes processed with BaselineRemoval and gaussianfilter_1d methods. Fourier filtering was performed by fitting Lorentzian functions to FFT peaks and analytically taking their inverse Fourier transform:

$$S \propto \mathcal{F}^{-1} \left[\left((\hat{k} - \text{Re } k)^2 + (\text{Im } k)^2 \right)^{-1} \right] \propto e^{-i(\text{Re } k)x} e^{-(\text{Im } k)|x|} \quad (\text{M5})$$

We see that the propagation length L_p , defined as the distance x until the real-space amplitude drops to 1/e of its initial value, can thus be estimated from the half-width at half-maximum $\text{Im } k$ of the Lorentzian peak simply as $L_p = 1/\text{Im } k$.

5. In Fig.4d, the calculated propagation length gives a value of 1 μm , while the authors claim that the HEP propagates for $\sim 1 \mu\text{m}$ in Fig.3b. How did they extract the experimental results of propagation length?

The quantitative propagation length (referring to distance until 1/e decay) is actually 0.5 μm . To extract this, we employed a similar analysis to the one discussed in the supplementary information of *Nature Photonics* **11** (2017) 350-360, which looks at the linewidth of the Fourier transform peak of the experimental line profile. The “propagation length” mentioned originally referred qualitatively to the distance when the filtered profile decays away to almost nothing. We replaced the original sentence with a more precise statement of the propagation length. A detailed description of the analysis was added to the same Methods section pasted above. Looking at Fig. R4, 0.5 μm is consistent with the quantitative calculation at $E=1.376 \text{ eV}$.

Figure R4: Hyperbolic exciton polariton propagation length L_p calculated from far-field optical constants with (orange) and without X^* (gray). Dashed lines indicate the calculated $L_p \approx 0.5 \mu\text{m}$ at $E = 1.376 \text{ eV}$, consistent with measurements.

REVIEWERS' COMMENTS

Reviewer #1 (Remarks to the Author):

I appreciate the authors for responding to my questions in detail. Unfortunately the authors mentioned that the data quality could not be further improved. On the other hand, I do recognize the advances and difficulties of their near-field technique and the novelty of the material system. Therefore I would recommend its publication in Nature Communications.

Reviewer #3 (Remarks to the Author):

The authors have successfully addressed reviewers' comments and the manuscript has been substantially improved. I can recommend its publication in Nature Communications.